# Capturing Russian drinking patterns with the Alcohol Use Disorders Identification Test: An exploratory interview study in primary healthcare and narcology centers in Moscow

**Maria Neufeld**[1]*, **Carina Ferreira-Borges**[1], **Anna Bunova**[2], **Boris Gornyi**[2], **Eugenia Fadeeva**[3], **Evgenia Koshkina**[4], **Alexey Nadezhdin**[4], **Elena Tetenova**[4], **Melita Vujnovic**[5], **Elena Yurasova**[5], **Jürgen Rehm**[6,7,8,9,10,11,12,13]

**1** WHO European Office for Prevention and Control of Noncommunicable Diseases, Moscow, Russian Federation, **2** Department of Primary Prevention of Chronic Non-Communicable Diseases in the Healthcare System, National Research Center for Therapy and Preventive Medicine of the Ministry of Health of the Russian Federation, Moscow, Russian Federation, **3** National Research Centre on Addictions – branch, V. Serbsky National Medical Research Centre for Psychiatry and Narcology of the Ministry of Health of the Russian Federation, Moscow, Russian Federation, **4** Moscow Research and Practical Centre for Narcology of the Department of Public Health, Moscow, Russian Federation, **5** WHO Country Office in the Russian Federation, Moscow, Russian Federation, **6** Institute for Mental Health Policy Research, Centre for Addiction and Mental Health (CAMH) Toronto, Ontario, Canada, **7** Campbell Family Mental Health Research Institute, Toronto, Ontario, Canada, **8** Institute of Medical Science (IMS), University of Toronto, Toronto, Ontario, Canada, **9** Department of Psychiatry, University of Toronto, Toronto, Ontario, Canada, **10** Dalla Lana School of Public Health, University of Toronto, Toronto, Ontario, Canada, **11** Institute for Clinical Psychology and Psychotherapy, TU Dresden, Dresden, Germany, **12** I.M. Sechenov First Moscow State Medical University (Sechenov University), Moscow, Russian Federation, **13** Zentrum für Interdisziplinäre Suchtforschung (ZIS), Universitätsklinikum Hamburg-Eppendorf, Hamburg, Germany

* neufeld.maria@gmail.com

## Abstract

### Background

Despite a considerable reduction in alcohol consumption, Russia has one of the highest levels of alcohol-attributable burden of disease worldwide due to heavy episodic drinking patterns. Further improvement of alcohol control measures, including early provision of screening and brief interventions (SBI), is needed. The legislative framework for delivering SBI in Russia was introduced in 2013. As part of the creation and validation of a Russian version of the Alcohol Use Disorders Identification Test (AUDIT), the present contribution explored challenges in using the AUDIT in Russia to inform a subsequent validation study of the tool.

### Methods

Qualitative in-depth expert interviews with patients and healthcare professionals from four primary healthcare and narcology facilities in Moscow. A total of 25 patients were interviewed, 9 from a preventive medicine hospital, 8 from a polyclinic, and 9 from narcology clinics. Also, 12 healthcare professionals were interviewed, 5 of whom were primary healthcare doctors and 7 were narcologists.

**Data Availability Statement:** \*\*\*PA@ACCEPT: Please confirm DAS w AU\*\*\* All relevant data are

within the manuscript and its Supporting information files.

**Funding:** This work was supported through a grant of the Russian Government to the WHO European Office for the Prevention and Control of Noncommunicable Diseases. This funding source had no role in the design of the study, its execution, data collection and analyses, interpretation of the data, writing of the manuscript or decision to submit it.

**Competing interests:** The authors have no competing interests.

## Results

Patients and healthcare professionals expressed difficulties in dealing with the concept of a "standard drink" in the AUDIT, which is not used in Russia. Various patients struggled with understanding the meaning of "one drinking occasion" on the test, mainly because Russian drinking patterns center around festivities and special occasions with prolonged alcohol intake. Narcology patients had specific difficulties because many of them experienced *zapoi*–a dynamic drinking pattern with heavy use and a withdrawal from social life, followed by prolonged periods of abstinence. Surrogate alcohol use was described as a common marker of alcohol dependence in Russia, not accounted for in the AUDIT.

## Conclusions

The provided analyses on the perception of the Russian AUDIT in different patient and professional groups suggest that a series of amendments in the test should be considered to capture the specific drinking pattern and its potential harms.

## Background

Developed by the World Health Organization (WHO) as part of a large international collaborative project, the Alcohol Use Disorders Identification Test (AUDIT) is a simple method of screening for hazardous and harmful drinking [1, 2]. Today, it is one of the most frequently used alcohol screening tests in the world, at both the individual and population levels [3–6]. Its main purpose is to detect drinkers at risk and offer them brief interventions at the level of primary healthcare (PHC). The overarching goal of this screening and brief intervention (SBI) approach is to identify at-risk individuals before health and social problems become more pronounced and require intensive and specialized interventions [4, 7, 8]. Facilitating access to screening, brief interventions, and treatment for alcohol use disorders has been included as one of the five most cost-effective interventions in the WHO-led SAFER initiative to reduce alcohol consumption and harm at the population level [9, 10].

In 2013, Russia introduced specific evidence-based screening procedures for risk factors in PHC facilities, including screening for "risk of harmful alcohol use", using first the CAGE screening tool (the name of which is an acronym of its four questions) and then the AUDIT [11, 12]. However, the existence of more than 60 different Russian translations of the instrument and the absence of any validation studies in Russian PHC facilities [13, 14] makes its use at the population level very difficult. In order to initiate a formal process of translation, adaptation and validation of the AUDIT in the Russian language and in PHC facilities in the Russian Federation, the Russian Ministry of Health and the WHO Country Office in the Russian Federation with technical support of the WHO European Office for the Prevention and Control of NCDs have formed a group of international and national experts, the RUS-AUDIT Project Advisory Board, to develop a study protocol for all the necessary steps [15]. As already outlined above, the experts were not only confronted with the different translations of the instrument, but also expressed a concern that the AUDIT might not capture heavy drinking occasions well, especially with strong spirits, typically found in Russia as well as various other Eastern European countries, which have a strong detrimental impact on health [16–18]. Specifically, there was a concern that the AUDIT might not correctly classify people with alcohol dependence, whose drinking patterns are characterized by prolonged periods of

heavy alcohol intake (the so-called Russian "*zapoi*") combined with alternating prolonged periods of abstinence. In international literature, the *zapoi* is operationally defined as a period of two or more days of continuous drunkenness when the person is withdrawn from normal social life [19, 20]. The Russian clinical guidelines on the prevention, diagnosis, and treatment of alcohol use disorders (AUDs) feature *zapoi* as a specific form of alcohol dependence and as part of "second-stage of alcoholism." There, the *zapoi* is defined as a dynamic drinking pattern of regular alcohol intake over several days or weeks (the actual *zapoi*), followed by abstinent periods of 7–10 days, up to several weeks or months, followed again by the *zapoi* episode. The *zapoi* episodes are accompanied by a pronounced withdrawal syndrome and other psychological symptoms such as insomnia, low mood, anxiety, feelings of guilt and so forth [21].

Another voiced issue in relation to screening for AUDs in the Russian context was that the AUDIT might not account for the combination of the *zapoi* pattern and the consumption of highly concentrated surrogate alcohols and other unrecorded alcoholic products. Unrecorded is a broad umbrella term for alcohol which is not accounted for in official statistics on alcohol taxation or sales because it is usually produced, distributed, and sold outside the formal channels which are under governmental control [22]. Unrecorded alcohol includes different sub-types: homemade alcohol, illegally produced and/or undeclared alcohol to evade excise tax, including counterfeits of alcoholic beverages, smuggled alcohol and alcohol brought legally over the border but recorded in another jurisdiction, and finally alcohol surrogates: alcoholic products that are officially not intended for use as alcoholic beverages but that are used as such, e.g. hand disinfectants, colognes, medicinal lotions etc. In Russia and neighboring countries, consumption of surrogate alcohol is persistently linked to severe forms of alcohol dependence in people from lower socio-economic strata, who can no longer afford beverage alcohol and are consuming surrogates in times when no other alcohol is affordable or available [23–27]. Although some important measures against surrogates were taken after the mass methanol poisonings in the Russian city of Irkutsk in 2016, certain types or surrogates remain available in Russian cities to this day, possibly fueled by the ongoing COVID-19 pandemic and the increased production of hand disinfectants [28, 29]. Overall, it is estimated that roughly one third of the total amount of alcohol consumed in Russia is unrecorded and that out of all of the unrecorded alcohol drunk, about 20% is consumed in the form of surrogates [22, 30]. Since the first three AUDIT consumption items assess the use of alcoholic beverages and do not specifically screen for the consumption of surrogates, Russian experts feared that the test might not perform well in identifying AUDs in Russia.

However, the main concern of the AUDIT application in Russia was that the concept of a "standard drink" that is used in the second test item. The concept, which is not frequently used in Russia and is hardly known in the general population or by health professionals in PHC, has been outlined by several research and practitioners groups in the past [31, 32].

The present study explored the perception of the AUDIT and general drinking behaviors in patients and healthcare professionals from Russian PHC as well as specialized alcohol and drug treatment services, i.e. the Russian narcology services (a special sub-discipline of psychiatry in Russia and some other countries of the former Soviet Union). By using qualitative in-depth interviews and content analysis, the present contribution provides a deeper understanding of Russia-specific drinking patterns and the difficulties of the AUDIT to capture these as part of the screening process. The outcomes of this exercise also informed the Russia-specific adaptation of the AUDIT as the end result, which was then formally validated in various PHC facilities in Russia, following a formal protocol [15, 33].

## Methods

### Aims and objectives

The objectives of this exploratory interview study were to identify any major problems in administering the AUDIT as a screening instrument for alcohol use in the Russian language, in Russian healthcare facilities, as well as any specific difficulties encountered during the assessment process and potential solutions for them. A special emphasis was put on the first three consumption items of the test as well as an overall exploration of potential Russia-specific markers for problem drinking and AUDs, which can be considered when adapting the AUDIT for use in Russian PHC facilities. The study had the following aims: 1) to explore problems in the understanding of the current AUDIT items by both–patients and healthcare providers, 2) to find ways of improving clarity of any AUDIT items causing difficulties with comprehension in patients and healthcare providers, 3) to explore possible solutions and existing practices of screening and quantification of risky use, mainly aiming at healthcare providers and 4) to integrate experiences about the use of the AUDIT from prior studies, including not-yet-published studies, and applications (healthcare providers only). Therefore, the interview guide probed the following areas: 1) the perception of frequency-volume questions and their importance (AUDIT-C), 2) the perception of different versions of the AUDIT, namely two distinct Russian translations featuring either the concept of a standard drink or a predefined frequency-volume table of the most commonly used beverages in Russia, 3) the specificity of Russian drinking patterns, most importantly single occasions of drinking where intoxication is reached as well as specific markers for problematic alcohol consumption in Russia, including consumption of surrogate alcohol (for the interview guide, please see S1 Appendix). Although all areas were covered in each of the conducted interviews, the interview guide was applied in a flexible manner in order to permit discussion of other relevant topics raised by participants.

### Study setting, design, recruitment, and participants

The study took place between the December 10, 2018 and February 6, 2019 in four different healthcare facilities in Moscow, Russian Federation.

A total of 25 patients from a general and preventive care hospital, a polyclinic (the main providers of PHC services in Russia), and two narcology centers were recruited and interviewed by the first author following a convenient sampling method, where patients that were present at the day of the interview in the facilities were approached by their treating healthcare workers and asked to participate in the study. A total of 12 healthcare professionals from the same facilities, five from the general and preventive care hospital and seven from the narcological centers, were recruited also following a convenient sampling approach. No more interview partners were recruited once data saturation was reached and no new key themes were identified in the interviews. The interviewer had received training in different interview methods, including interviews in clinical practice and research.

Since this study's main goal was one of in-depth understanding of the AUDIT and quality improvement of screening procedures, without a need to collect any identifying patient information, it was considered to be part of routine care by participating primary healthcare institutions, which obtained formal approvals from local authorities. For specialized narcology centres, where no screening procedures are carried out as part of routine care, the study underwent ethical review and was approved by the Ethics Committee of the V. Serbsky Federal Medical Research Centre of Psychiatry and Narcology of the Ministry of Health of the Russian Federation, Protocol No 25/6 from 21 January 2019. Recruitment and interviewing of health

professionals and patients from PHC facilities took place prior to the recruitment of the nar-cology sample, while the narcology samples were recruited once ethical approval was obtained. We asked for verbal consent from all health professionals and patients willing to participate. The ethical approval subsequently obtained from the V. Serbsky Federal Medical Research Centre of Psychiatry and Narcology covered the full study "Adaptation and validation of the Russian-language version of the AUDIT (Alcohol Use Disorders Identification Test)". All parts of the study were fully compliant with ethical principles, including the provisions of the World Medical Association Declaration of Helsinki, as amended by the 59th General Assembly, Seoul, the Republic of Korea.

## Procedures

After explaining the main objectives of the study via a standardized script and obtaining verbal consent from the patients, qualitative semi-structured expert interviews on the perception of the AUDIT were conducted one on one in a private room and audiotaped. The interviewer had no access to information that could identify individual participants during or after data collection. Expert interviews are a standard method of qualitative approach and they denote in-depth interviews with a qualified specialist or expert on a particular topic [34]. The previously rigid hierarchy between formally acknowledged experts and lay people tends to give way to more flexible and situational structures of interaction, which is why both—PHC and narcology health professionals and patients—were defined as experts in the framework of this study [35]. Narcology specialists and patients were conceptualized as experts because of their daily experiences with AUDs in the Russian context, including the aforementioned specificities of Russian drinking patterns, and PHC specialists and patients were defined as such since they are the target audience for the AUDIT. The perception of the AUDIT needed to inform the adaptation process, and it was crucial that the final Russian version of the test was intelligible to PHC patients and health personnels and was easy to navigate.

## Data collection and analysis

For the qualitative analysis, audio recordings were transcribed and translated into English and imported into qualitative data management software (MAXQDA, Version 12.3.4) [36, 37]. The interview transcripts underwent an inductive thematic analysis and key themes of the interviews were identified and subsequently coded into categories. An original code list was based on the newly emerged themes as well as the original key themes of the interview guide. Coding was then conducted by the first and last authors, who also cross-referenced the codes to ensure that a consensus in the understanding of the data was reached. The themes were included in final analyses when they were introduced or discussed by numerous participants, and, in the following, selected illustrative samples from both patient and healthcare profession-als are featured to narratively represent the analyzed key themes.

## Results

A total of 25 patients gave consent and were interviewed as part of the study. Out of them, nine patients (five men and four women) were recruited from a general and preventive care hospital, eight (3 men and 5 women) from an outpatient general polyclinic, who presented to a general practitioner and nine(3 men and 5 women) from inpatient narcological facilities (all narcological patients were treated for alcohol-related conditions). There were slightly more women (14 out of 25) in the patient sample and the age range was broad, spanning from 19 to 85 years. The sub-sample of narcology patients was considerably younger (mean age: 41.7, median age 38) than the sub-sample of patients from the general and preventive care hospital

(mean age: 62.6, median age: 71) and the polyclinic (mean age: 57.1, median age 56) and a narrower age range of 31–63 years, as compared to 19–85 and 31–81 for the two other facilities, respectively. Moreover, the narcology patients have reported a lower age at drinking onset than the other two subsamples of patients, with mean age at drinking onset of 16.4 years for the narcology patients and the mean age at drinking onset of 18.1 years for the hospital subsample and 17.5 years for the polyclinic sample.

A total of 12 healthcare workers have agreed to be interviewed, five of whom were preventive and healthcare professionals (four cardiologist and one preventive medicine specialist by training) and seven narcology specialists (one of whom was also initially trained as a general practioner). For healthcare workers, no further demographic characteristics are provided to maintain their anonymity.

In general, all interviewed patients and healthcare professionals agreed that questions about both frequency of drinking and the maximum amount of alcohol consumed at once (during once occasion) could indicate if a person had any potential problems with alcohol. In the following, we explore the identified issues in the perception of the first three AUDIT consumption items as well as further identified key themes that emerged from the interviews.

Overall, health professionals emphasized the need for a screening instrument like the AUDIT in the Russian Federation and some narcology specialists reported positive experience in applying the AUDIT in their line of work as part of assessment procedures of patients.

## AUDIT item 1: Frequency of drinking

The first item of the AUDIT assessed frequency of drinking (for a complete back-translated version, please see S2 Appendix).

*AUDIT item 1, as used in the interview guide, back translated into English.*

1. How often do you drink alcoholic beverages?

(0) never

(1) once a month

(2) once a week

(3) 2–3 times a week

(4) 4 or more times a week

Overall, PHC patients had no issues with this test item, most of them reporting that they drank irregularly and only during special occasions, such as celebrations of certain holidays.

### PHC_Patient_11 [Female, 56 years old, age at drinking onset: 15, current drinker]

*Well, in general I don't drink. Only if there is some kind of holiday, there is something like a wedding or a birthday celebration. [. . .] So overall, it maybe 100 grams or so. Well, maybe 150 grams [. . .] So this is at big events, at weddings or maybe at anniversaries. [. . .] A shot of vodka, yes. Also, I could have a little wine on holidays. [. . .] How much is a shot glass? Well, maybe 100 grams. But there are also smaller shot glasses. [. . .] So maybe 100 grams during a*

*whole evening, for the whole holiday. Well, a wedding happens once, twice a year and besides this, I do not drink at all.*

Some narcology patients clarified that they were now sober and in treatment and stated that before being admitted to the narcology clinic, they were either abstinent for a while or had experienced a *zapoi* episode within the last 12 months, or both, which made it difficult for them to give a definitive answer to the first item.

The first AUDIT test item was generally comprehensive and not problematic for patients, which awas confirmed by the accounts from health professionals.

### PHC_Professional_2

*Usually people understand [. . .] Well, as far as frequency is concerned, there are no questions in general [. . .] Questions arise usually in relation to what to do if they drink different alcoholic beverages. . .well, in different quantities. How to average that amount to have the number of standard drinks? Many say, for example, that one day they can drink five glasses of wine, and another time they will drink one glass of wine–so what are they supposed to answer in this case, how to calculate the average amount? [. . .] Usually we ask what is more typical for them, that is, what situation is more frequent. Something will be considered as a more typical and something as a less typical behavior for a person. And so they have to choose a more typical option.*

## AUDIT item 2: Typical quantity

The second AUDIT item assessed quantity of intake on a typical drinking occasion.

When asked in an open format, all participants were generally able to report on their usual intake, describing the consumed volume either in milliliters, grams, or glasses of a particular consumed beverage. Most PHC patients reported rare occasions of drinking, sometimes heavy drinking, which centered around special occasions and festivities as well as weekends. Some patients mentioned that it was difficult for them to report on how much they usually drink because this would always depend on the specific drinking occasion, but did not experience any major difficulties when asked by the interviewer to pick a situation which was more typical for them personally.

To explore the intelligibility of standard drinks, patients were presented with different AUDIT versions: one version that used the concept of standard drinks (back-translated as "doses" into English) and one where the standard drinks were entirely dropped and replaced with a table of alcoholic beverages volumes, which translated directly into AUDIT scores). Both versions were taken as the two alternate Russian translations of the AUDIT from the official website of one of the original developers of the screening tool [38, 39]. Patients were shown the different versions in an alternating order and were asked for their feedback.

*AUDIT item 2, as used in the interview guide, first version that uses the notion of standard drinks, back translated into English. Source: [38].*

2. How many doses do you usually drink when you drink?

(0) 1–2

(1) 3–4

(2) 5–6

(3) 7–9

(4) 10 or more

*AUDIT item 2, as used in the interview guide, second alternate version that uses a frequency-volume table, back translated into English. Source: [39]. The original version omits the ml in the cell "dry wine" and does not feature 9 standard drinks in the cell "7 or 8", which seems to be a conceptual mistake.*

| | **What is your typical alcoholic beverage intake on the day of drinking?** | | | | |
|---|---|---|---|---|---|
| | Standard portion | Vodka (ml) 40 vol% | Fortified wine (ml) 17–20 vol% | Dry wine 11–13 vol% | Beer (bottle) 5 vol% |
| (0) | 1 or 2 | 30–60 | 75–150 | 75–150 | 250ml-1 bottle |
| (1) | 3 or 4 | 90–120 | 225–300 | 300–400 | 1.5–2 bottles |
| (2) | 5 or 6 | 150–180 | 375–450 | 500–600 | 2.5–3 bottles |
| (3) | 7 or 8 | 210–240 | 525–600 | 700–800 | 3.5–4 bottles |
| (4) | 10 or more | 300 and more | 750 and more | 1000 and more | 5 and more bottles |

The subsequent reflections of patients and healthcare specialists on the concept of standard drinks and their intelligibility in the Russian context represents one of the key themes of the entire analysis.

Unsurprisingly, the version of the AUDIT 2 items that operates with standard drinks was much more difficult to understand for both PHC and narcology patients, and the latter group had the greatest difficulties. Both patient groups stated that they needed help and/or at least more time to understand what is required from them when replying to the standard drink version of the test item.

**Narcology_Patient_9 [Male, 63, age at drinking onset: 16, currently in treatment]**

*This is difficult to convert. I even saw such a questionnaire in a polyclinic once and there, you simply don't have the time for this. You stand at the reception desk and fill out all these forms, a lot of forms, and there, you don't have the time to think about all these standard drinks and glasses.. [. . .] I think you should not ask this question as based on ethanol units. It should be in glasses. Still, everyone understand this–small and big glasses.*

**Narcology_Patient_4 [Female, 38 years old, age at drinking onset: 17, currently in treatment]**

*I don't understand this. Not clear. [So from your point of view, should we be asking in bottles then or what?] Most likely so, yes.*

Some healthcare workers emphasized that one of the main differences between Russian and Western European drinking patterns is the lack of understanding what a standard drink is in the Russian context, which affects the entire screening process.

### Narcology_doctor_4

*In Russia, the concept of a standard drink is generally very relative. Because as I understand, in the West, people can count in drinks, because they often drink alcohol in a bar and it is easier to count. In our culture, people more often consume alcohol at home. When they drink at home with others, it's harder for them to count. And it was necessary to adapt the drinks that are indicated in the test, to translate along with the patients, to calculate how much standard drinks they actually consume.*

Both narcology and PHC doctors emphasized that a thorough explanation and preferably a visualization of the standard drink is also needed in the self-administered version of the AUDIT, which has to be filled out by the patients without the assistance of a healthcare worker. Overall, there seemed to be a consensus that administering the AUDIT as part of an interview with the healthcare professional is the preferred option because concepts like the standard drink could then be explained to the patient during the interview.

When presented with an AUDIT version which included a frequency-volume conversion table instead of standard drinks, respondents stated that they could navigate the question and the different answer options much more easily and understood what kind of response was required.

### Narcology_Patient_2 [Female, 33 years old, age at drinking onset: 16, currently in treatment]

*So, if you present this as standard drinks–it's not comprehensive for me. But if you present it like this, this is very comprehensive. [So, in principle with this show card here you can manage?]. Yes, of course, if you have this, then this is clear. [So, you could manage this test on your own if you had the show card?] Yes!*

### Narcology_Patient_1 [Female, 34 years old, age at drinking onset: 17, currently in treatment

*I think people will look more at these numbers here [the table with volumes]. Because the 10 grams- no one will calculate this. I think if you show bottles here or glasses, it will make it much easier to answer this question. Because I would not understand what this means: three to four and five to six standard drinks. Because it's not clear what you mean: either litres or grams or. . . [So in either case we need some form of table or showcard or. . .?] Yes, some way of correcting this. Because people won't understand this. Or you won't understand our people.*

In fact, some respondents even suggested introducing such a conversion table or a show card themselves to make the test intelligible to other patients.

### PHC_Patient_15 [Male, 72 years old, age at drinking onset: 17, has not drunk for the last 3 years]

*This should be translated into normal language. I myself understand how many 10 grams of ethanol are. I remember how much alcohol is there and how much vodka is, I myself understand this. But it is necessary to translate this for everyone because one cannon figure it out immediately. Well, it is better to say something like. . . . A wine glass. 50 grams of vodka. Then*

*everything will be clear. And if there will be 10 grams of ethanol instead, then one needs to think. [So, we need to explain right away that we are talking about a glass of vodka, a glass of wine and so forth. . .] Yes, it will be faster to answer like this*!

The usefulness and value of a conversion table that translated the consumed volumes of alcoholic beverages into standard drinks or even directly into AUDIT scores was also emphasized by the interviewed PHC doctors, who were working with the AUDIT-C as part of a research study. The reflections of PHC specialists and narcologists support the statements of patients that an assistive device, such as a table or a show card displaying the most commonly consumed beverage types, is needed to ensure comprehension.

**Narcology_doctor_3**

*Neither in a professional environment nor among consumers of alcoholic beverages- we simply do not have the concept of a standard drink, it is very underdeveloped. Standard drink is a very complex concept. Often, when you ask a question, the frequency is clear, but the amount —there the main questions arise. We had special flashcards, which were in color and laminated, which displayed certain volumes of alcohol: "This is one drink. This is beer, alcohol. How much do you drink? Let's count!" People do not understand. And in general, the problem of the standard drink, it is big one*

Both narcology and PHC specialists, emphasized that standard drinks are not known and commonly understood in their respective healthcare settings, which makes it necessary to invest time into either explaining the standard drink concept to the screened patients or to invest more time in the training of healthcare professionals who carry out the AUDIT screening, and make sure that they are able to correctly transform beverage volumes into standard drinks.

At the same time, a PHC specialist emphasized the educational function that standard drinks could have as part of an intervention procedure.

**PHC_doctor_4**

*Still, the concept of a standard drink is needed. It must be kept in the question and explained to the patient. Firstly, for what it is. So, this concept has also an educational function. Many people believe that low alcohol drinks, for example, are better than spirits. They believe that they contain less alcohol, so to speak. That is, they do not understand that the main component of an alcoholic beverage is ethyl alcohol and that it is important for us to understand the total amount of ethanol consumed. And they should understand this. [. . .]. This is why the concept of a standard unit was introduced in the first place.*

The conversation between a patient and a healthcare professional on what a standard drink is could also be considered as a link between the screening component and the actual brief intervention. It can serve as an entry point for conversation on alcohol consumption and risks, explaining to the patient that consumption of different alcoholic beverages, including low-alcohol drinks, will add up to the same amount of standard drinks as the consumption of a single beverage.

Besides the highlighted suggestions to use milliliters as a familiar and comprehensive unit of measurement, several middle-aged patients used grams when reporting their alcohol use and one patient from the narcology sample even insisted on using grams to describe Russia-specific drinking patterns. Moreover, some narcology patients stated that they felt that the

entire test item was useless for them because the amount they would consume on a typical day of drinking would by far exceed the suggested volumes, highlighting that narcology patients are not the correct target audience for a screening test like the AUDIT.

**Narcology_Patient_1 [Female, 34 years old, age at drinking onset: 17, currently in treatment]**

*The thing is if you ask alcoholics, you don't even need to show them this [conversion table]. Every alcoholic will tell you it's more than 10 [standard drinks].*

## AUDIT item 3: Frequency of heavy drinking

The third AUDIT item on frequency of heavy drinking occasions was asked in an open-ended format to avoid using the concept of standard drinks or the pre-defined table of alcoholic beverages on the second test item (see S3 Appendix). Instead, frequency of heavy drinking occasions in the past month was assessed using two open-ended questions, the first of which asked for the maximum intake on one occasion, which could be defined as a 24 -hour period if the person had trouble understanding what one occasion meant, and the second follow-up question asked how often these situations had occurred in the past 30 days.

> *Alternative AUDIT item 3, as used in the interview guide, to explore maximum intake quantity.*
>
> How much alcohol did you drink when you consumed the most during one occasion [defined as 24 hours]? [Please indicate in bottles of beer, wine or vodka?]
>
> How often did you consume this amount of alcohol in the course of the last 30 days?

When responding to this question, both, PHC and narcology patients reported their maximum intake not only in bottles as instructed, but also in glasses, liters, milliliters and sometimes grams of alcoholic beverages. None of the interviewees came back to the concept of standard drinks as was used in the previously discussed test item.

Another key theme that emerged in relation to this test item was the understanding of the concept of "one setting" or "one drinking occasion." Patients and healthcare professionals alike described alcohol use as a mundane phenomenon in their lives and as an important part of social interactions with friends and family. Although drinking occasions were described as something that does not happen often and is rather linked to special events like celebrations, these events would typically last for several hours and even days during which people could get very intoxicated. For instance, wedding celebrations that would last for several days were typically brought up as an example of the latter.

The social environment and peer group and the specific situation were reported as key factors that would strongly shape the drinking behavior, along with specific alcohol types consumed. Moreover, the maximum alcohol intake and whether someone would reach intoxication during these social gatherings, but also when consuming alcohol alone, would also depend on the employment situation, i.e. whether being intoxicated was tolerated at their workplace or not.

Therefore, it was sometimes difficult for patients to indicate how much they would drink during one occasion, and when reporting the maximum intake they would often specify that this was during a longer period of several hours, an entire day or longer, for instance over a weekend.

**PHC_Patient_17 [Female, 72 years old, age at drinking onset: 20, drinks from time to time but much less than earlier]**

*I could drink a bottle of dry wine. It was probably six months ago, at the dacha. Well, this is in the process, for several hours, not at once. Not so that I sat down and just knocked myself out. No, it was in the process of having a long conversation, while socializing.*

**PHC_Patient_6 [Male, 61 years old, age at drinking onset: 30, did not consume alcohol for the last 15 years]**

*Well 150, 200 grams. [Sprits?] Spirits. [On one occasion?] In one day, not at once. On one occasion is a lot.*

Therefore, most participants—PHC and narcology patients—but narcology patients even moreso, found it much easier to refer to a clearly defined period of 24 hours when reporting their maximum intake of drinking. When offered this timeframe rather than "one occasion", most respondents found it easier to navigate and recall their maximum intake. Subsequently, when asked specifically for their maximum intake within 24 hours, patients could describe the consumed volume much more easily and the test item worked much better for narcology patients in particular.

**Narcology_Patient_2 [Female, 33 years old, age at drinking onset: 16, currently in treatment]**

*So, I could drink about five to six litres [of beer] during 24 hours. [. . .] [Would it somehow influence your answer if we were asking how much you drank per occasion?] It would! I think it's better how much you drink per day. [Per day or per 24 hours? What do you think?] Per 24 hours. Some people work shifts. Like a two shifts rotational pattern. Some people have regular 5 working day weeks, some people work shifts. So, I think it's better to ask for 24 hours. Because it can be that someone is working the shift and doesn't drink but then has two days off and is drinking the first 24 hours and then is trying to make themselves presentable the next 24 hours before getting back to work.*

Although PHC specialists did not report many issues with the "one drinking occasion" concept, some reflected on the difficulties of capturing the drinking patterns of heavy drinkers in Russia, mentioning that the ambiguity of this test item leaves a lot of room for personal interpretation, which is not the goal of a standardized test.

Correspondingly, narcology specialists pointed out that it would be much clearer to use the concept of maximum intake per 24 hours because this would eliminate the ambiguity and allow for standardized assessment of different drinking scenarios. It was also emphasized that this concept is used in the narcology setting.

**Narcology_doctor_5**

*The problem with that is if you ask someone in primary healthcare 'What is your maximum alcohol intake per occasion' they will tell you something like '100 grams'. But the problem is that this way, you won't get the person's real maximum intake. Because maybe he had these*

*100 grams about three times or four times. So, this is not valid then. It's better to assess this like we do this in narcology. To ask 'How much do you drink in 24 hours'?*

Naturally, narcology patients reported much higher volumes of consumed alcohol overall for 24 hours than PHC patients and they had no difficulties in understanding and navigating this question as they seemed to know exactly what their maximum intake was.

## Further indicators of problem drinking that are specific for Russia

The last part of the interview guide was dedicated to the exploration of any additional indicators and potential markers of problematic drinking in Russia that could be accounted for in the construction and validation of the Russian AUDIT. Interview partners were asked to reflect on a set of indicators, such as having one or more episodes of *zapoi* in the past year, occurrence of excessive drunkenness, hangover, or going to sleep at night clothed because of being drunk as well as consumption of surrogate alcohols (see interview guide in the S1 Appendix for more details).

Overall, all the suggested behaviors were perceived markers for problem drinking. Excessive drunkenness, *zapoi*, and loss of control, and, probably most importantly, consumption of surrogate alcohol where all perceived to be signs of AUDs by patients and healthcare workers alike.

When reporting their maximum alcohol intake, various narcology patients automatically reflected on their own *zapoi* patterns and described in greater detail how these drinking patterns were different from drinking patterns of people who do not have an AUD and how this would also affect the screening procedures for potential AUDs.

### Narcology_Patient_1 [Female, 35 years old, age at drinking onset: 17, currently in treatment

*When I start to drink this means that I start to drink. Because these are different things for an alcoholic and for a person who is not an alcoholic. We [alcoholics] would understand this in our own way, while people who are not alcoholics will understand this in completely different way. They would tell you that they drink once a month, like for instance a couple of glasses during a birthday celebration. But for us, this is more about drinking. Like really drinking, when you start to drink and have a zapoi and so forth and this is different. [. . .] So again, this and the other question [on number of standard drinks] are not correct for alcoholics. Now, all the people who are in here for treatment, they will just tell you that they drank the maximum amount and that's it.*

Many of these accounts illustrate the spiraling dynamic of *zapois*, where the individuals would drink heavily for several days, then slowly decrease their daily intake to a minimum because of the strong intoxication and withdrawal syndrome, then get sober and relapse again.

### Narcology_Patient_3 [Female, 32 years old, age at drinking onset: 14, currently in treatment]

*The longest zapoi I have ever had was a month. But it doesn't mean that I drank two litres per day. It was only at the beginning, the first day that I could drink two litres. Then it becomes less and less [. . .] Still, it is a zapoi. Every day you get up and you need to drink to ease the hangover. So during the day you hangover-drink("опохмелиться"/opokhmelit'sya") but the dose gets lower every day. From two liters to less. But still, this is a zapoi. [. . .] So, a zapoi is" to drink and to sleep. To drink and to sleep. This is the main thing. Sometimes some action is*

*going on, for instance the crowd moves from one apartment to another [. . .] you also exchange one crowd for another until you are completely out of it and when it's about nighttime you sleep. Then you get up, look for money. If you have money, you don't have to look for anything, just buy [alcohol] again, sit down and drink until you are piss drunk and then you sleep again. So, this is what it is: a bad zapoi. The last phase of alcoholism.*

**Narcology_Patient_2 [Female, 33 years old, age at drinking onset: 16, currently in treatment]**

*So last year I had a zapoi. More than two days, more than a week actually. I was drinking non-stop, you could say around the clock. So, I bought a bottle of wine and thought that it's enough for me. But it wasn't enough, so I went on and bought another two bottles. I thought that I can just keep it around. But I couldn't. So, this is how this becomes a 24 hour thing. You wake up and you don't feel well. You drink and you sleep, drink and sleep, drink and sleep. And this went on for nine days. [. . .] So, for me this is what a zapoi. You wake up, you drink, you sleep and so on. For me this was nine days in a state of stupor. So, basically I don't remember these days at all. I just woke up, drank and went to sleep again. Just somehow automatically managed to feed my cat. I think it was some subconscious thing that you woke up, had to feed him and then- morning, evening and that's it. [. . .]. That's about dependence because a normal person won't drink this way. [. . .] So, when we talk about alcohol-dependent people, we're talking about at least two days [of drinking], even more.*

The idea of self-control and the notion of knowing when to stop as part of a *zapoi* was a recurring theme in many of the patients' accounts, not limited to the narcology patient sample.

**PHC_Patient_17 [Female, 72 years old, age at drinking onset: 20, drinks from time to time but much less than earlier]**

*If a person just drinks to relax, to get pleasure, even if he is in a zapoi, but he has his own boundaries, some kind of self-control. Not because of what my husband says to me or my uncle or aunt–this is about self-control. And when a person is already spiraling and cannot stop, drinks for a day or two and. . .. this is already alcoholism.*

At the same time, narcology specialists clearly highlighted that the main idea of a *zapoi* episode is precisely the loss of control over one's drinking. Therefore, a *zapoi* as a part of alcohol dependence syndrome is clearly distinguished from prolonged heavy drinking episodes that are mostly associated with holidays, celebrations, and leisure time in Russia.

**Narcology_doctor_6**

*When talking about a zapoi we are always talking about the loss of control. It's not about situations like for instance New Year, where I am visiting my relatives, we sit behind a table as a big family and of course- we drink there every day. Is this a zapoi? [. . .] For instance, when you have time off and you meet your friend [. . .] and you are going out enjoying your freedom and loosening up and drink every day, but you are not getting drunk. Is this a zapoi? So, for a narcologists a zapoi is always about control loss.*

Other drinking behaviors like excessive drunkenness, hangover, or going to sleep at night clothed because of being drunk were perceived in a more ambiguous way, with interview partners often stating that this is clearly a behavior that indicates a certain risk but not necessarily indicates an AUD because heavy drinking occurs also as part of festivities and celebrations.

**Narcology_Patient_2 [Female, 33 years old, age at drinking onset: 16, currently in treatment]**

*[Twice a week or more: occurrence of excessive drunkenness, hangover, or going to sleep at night clothed because of being drunk]. It's the risk zone. This is not yet addiction but this is absolutely a risk zone. Because I had this when I was not yet an addict. It was some kind of warning sign that I had to stop. [And what exactly? The excessive drunkenness the hangover or...] The excessive drunkenness! But this again depends on the beverage. So, for instance when I was drinking beer, I was absolutely capable to undress myself before going to sleep. But if this was strong alcohol, vodka, then yes...it happened that I once woke up still dressed because I passed out.*

Consumption of surrogate alcohols, on the other hand, was clearly perceived as an indicator for AUDs and represented another large key theme in the interviews. Both, PHC and narcology patients emphasized the strong link between surrogates and AUDs and claimed on several occasions that consumption of surrogate alcohol is a clear sign of alcohol dependence and social decline.

**Narcology_Patient_1 [Female, 35 years old, age at drinking onset: 17, currently in treatment**

*This is really clear. A person who is not an alcoholic won't drink colognes. We don't even have to discuss this one here. [So, this is quite a clear criterion for alcohol dependence?] Of course!*

**PHC_Patient 17 [Female, 72 years old, age at drinking onset: 20, drinks from time to time but much less than earlier]**

*[So, you think out of all listed criteria, this is the most serious one?]. Well, of course. You see, there's a person who drinks vodka—after all, vodka is a natural product, if it is not a counterfeit of course, if it's not illegal vodka. [...] And then there is a person who deliberately drinks cologne or some kind of cleaning agents, which do harm, colossal harm to one's health, then this is already degradation of one's personality. [...] This is already the next stage of alcoholism.*

**Narcology_Patient_8 [Male, 31 years old, age at drinking onset: 20, currently in treatment]**

*This is already death [...] because everyone knows that if you drink non-beverage alcohol you will get blind. [...] I've tried it once and I felt very, very bad afterwards. Worse than if you drink normal vodka. [...] It was a counterfeit, like a bottle of Hennessey whiskey that I've bought for 150 rubles. [...] Some people drink hawthorn lotion and [other medicinal tinctures], even some of my friends do and they are happy with this because they don't have the money for vodka. They get into withdrawal and they are happy that they have this. [So, would you say that these people have an alcohol problem?] I would say that they are goners.*

**Narcology_Patient_7 [Male, 45 years old, age at drinking onset: 14, currently in treatment]**

*This is already the bottom, the absolute rock bottom. [...] You really hit the rock bottom when you do this. [...] Some of our pharmacies, they sell pharmaceutical alcohol illegally. I think it has about 95%. So, in reality this alcohol is meant to be used as a hand disinfectant for the surgeon before the surgery. So, at some point, this one thought shot through my mind*

*'What the hell am I doing here? What am I drinking?' [. . .] So, I somehow realized [. . .] that I'm already at the brink of an abyss here.*

Moreover, several patients stressed that that consumption of surrogates in their respective context was not just a sign of alcohol dependence but of social drift and impoverishment because surrogates were the cheapest and the most frown-upon form of alcohol in their communities. Several accounts and statements highlight that consumption of surrogates is heavily stigmatized even among people with AUD and that only those who had no money at all for other types of alcoholic beverages were consuming surrogates.

**PHC_Patient_9 [Male, 55 years old, age at drinking onset: 16, did not consume alcohol for the last 15 years because of self-reported alcohol dependence]**

*I think this indicates the absence of money. So, for instance, if you have 1,000 rubles in your pocket, you can go to the store and buy normal vodka. But you can also buy windshield wiper fluid for 15 rubles if you only have 30 rubles. So, if you have the shakes and you need something to ease the hangover ("опохмелиться"/opokhmelit'sya"), you won't think long. You just go and buy it. [. . .] The question that you ask is whether surrogates are consumed because of alcoholism or not. Buy if you have money, you will buy normal things, so I think that the question is not really correct. Surrogates are consumed simply because of the lack of money. Yes, of course, this question is about alcoholics, about alcoholism. Yes, it's about that but I think it's not really just that because an alcoholic, if you have the money, you will go and buy other things. [..] If you have money, you will buy normal vodka and if you don't, you buy windshield wiper because you don't care. [So, if I understood you correctly, this is not about being an alcoholic but about being a poor alcoholic?] Yes! Yes, exactly this!*

**Narcology_Patient_2 [Female, 33 years old, age at drinking onset: 16, currently in treatment]**

*I never tried [surrogates] but you should ask this question [. . .] So, this is definitely about addiction when people come down to drinking surrogates. [. . .] This is really the rock bottom, not even our clinic here, but possibly some other clinic [. . .] for really lower social status people. I think people end up with surrogate alcohol when they no longer can afford alcohol for-. . .When they don't have enough money for a good quality beverage, they drink the surrogates. So, this means that the person is sick. It's not even dependence, it's sickness.*

**Narcology_Patient_9 [Male, 63, age at drinking onset: 16, currently in treatment]**

*A lot of people dilute colognes with water and drink them. Especially those, who come out of prison. [. . .] So, this is really a pointer and highlights alcohol dependence. I drank non-beverage alcohol because it was there and no other alcohol was available. So, I decided to try this. [So, because of availability?] Availability, yes. I was told that this is medicinal alcohol and I got a canister for something, I don't remember what, but I needed this for some reason and it was there [. . .] Of course, only a person with alcohol problems will drink non-beverage alcohol.*

The personal accounts of patients who encountered surrogate alcohol in their respective communities suggest that various forms of surrogate alcohols as well as other types of unrecorded alcohol, such as homemade alcoholic beverages and counterfeits, remain widely available despite various introduced measures, including the ban of certain alcoholic products after the mass methanol poisoning in Irkutsk. The Irkutsk poisoning was mentioned several times

by the patients and many of the interviewees referred to the issue of surrogate alcohol use as a big problem they were aware of, which is closely linked to overall enforcement issues of alcohol control policies in their communities.

**PHC_Patient 17 [Female, 72 years old, age at drinking onset: 20, drinks from time to time but much less than earlier]**

*This is just a disaster. Well, I saw it myself. They were buying these [surrogates] in pharmacies just in front of me. Do you remember this hawthorn situation [the Irkutsk poisoning]? Yes, there are many alcohol-containing products like this in the pharmacies. This is a terrible thing. This is scary when people are already starting to drink colognes. After all, they even drink cleaning agents. This is already degradation. It's even worse than alcoholism. This is already a degradation of personality. I personally think so.*

Overall, patients expressed a certain sense of fatalism when talking about the availability of surrogate alcohols and alcohol in general, emphasizing that there were different products coming from many sources.

**Narcology_Patient_3 [Female, 32 years old, age at drinking onset: 14, currently in treatment]**

*After my zapoi, when I was already feeling better, I just learned by chance that in Irkutsk 17 people died. Some counterfeited batch came out on purpose. I think they did this on purpose because of the competition. Because it's not only these small bottles. The thing is that alcohol is not sold after 11 PM anymore. And in our town they also sell this in the houses, in every second house they are selling either ethanol or samogon or they are giving you this like a credit. Some people accumulate debts of five thousands, three thousands, four thousands roubles. . . Different people, different debts. So, they give this to them, on credit and not. . .And it's cheaper than in the stores and you can buy this at night. [. . .] Now they sell this more from the houses. I no longer see these small bottles. [. . .].It did not disappear and yes, they find ways, they do.[So it's still there?] Ye,s it's still there and still works the same way. At the houses and apartments. People sell from their private homes. [..] There is no more hawthorn but all these other things, they stayed. [. . .] There is always a way to find it. Even more so in a small town, where everyone knows each other, where you know people, for instance you know that one saleswoman in this shop and she can sell it to you and just check this at the cash register in the morning. People find their way regardless. [. . .] Even looking at myself, I can tell you from own experience that If you really want it, you'll always find a way. [So even now, in the year 2019, regardless of all the alcohol control policies introduced, all the bans. . .] It's still all the same. At least in the villages and small towns. I can't say how things are in the cities. Maybe things are the same in small cities. [. . .] Colognes. People drink colognes. [. . .]. In my case this was because I ran out of money, I had nothing left and to get rid of the shakes, I needed something. And the cheapest thing for 25 rubles was [hawthorn lotion], this was the cheapest to find. Again, we are talking about a small town here, so you just go around scrounge money. 'Give me two roubles. Give me five roubles.' Once you managed to scrounge up enough money you go and buy it. This is because it's so cheap."*

Based on the perspectives provided by healthcare workers, there seemed to be a general consensus that consumption of surrogate alcohols indicates alcohol dependence against a background of low socioeconomic status.

## Discussion

The analyzed accounts of PHC and narcology doctors and patients highlight different aspects of Russian drinking patterns, which make the application of the AUDIT very challenging in the Russian context. There seems to be a clear understanding and consensus among all interview partners that using the concept of the standard drink in the Russian AUDIT does not work without providing any further explanations to healthcare workers or patients. The use of assistive devices such as tables or flashcards that would help with the translation of drinking volumes into standard drinks and the respective scores was brought up by several participants.

The interviews have overall confirmed the voiced concerns of experts that the AUDIT might not well capture the Nordic drinking pattern that is prevalent in Russia, which is characterized by heavy episodic drinking occasions with predominantly spirits [16–18]. Assessment of heavy episodic drinking and specifically the prolonged episodes of *zapoi* and the differentiation from "festive drinking", which centers around special occasions and holidays in Russia, also poses a specific challenge for the screening process. Overall, the distinctive concept of heavy episodic drinking as consuming a substantial amount of alcohol over a short period of time is not well understood in the Russian context because most of the drinking occasions are lasting for several hours as part of birthday, wedding, or other celebrations, where people reach different degrees of intoxication.

Moreover, the analyzed interviews highlight that drinking alcohol surrogates remains widespread in people with AUDs, which is a well-known and documented phenomenon in clinical populations in Russia and neighboring countries, including people in prisons [23–27, 40, 41].

Before moving on to the discussion of these findings in the context of the Russian AUDIT validation project [15, 33], limitations of this study need to be mentioned. As outlined, the results presented here are based on anonymous self-reports of patients and healthcare professionals that were recruited as part of a convenience sampling method, where the bias of the sample is unknown and cannot be measured and where results cannot be generalized to the larger population. Although we seemed to have reached sample saturation because no new key themes or specific topics were emerging after a while, the sample size can be considered as small, especially when accounting for the subsamples. Finally, the usual biases of self-report studies apply here, including social desirability or memory bias.

As such, the outlined results clearly indicate and confirm the need for an adapted version of the AUDIT, which would be able to capture the specificities of the described drinking patterns and have therefore informed validation project of the AUDIT in Russia. Based on the described outcomes of the study, the following changes were introduced into the now-validated Russian AUDIT: 1) a specific frequency-volume table with the depiction of the most commonly consumed alcoholic beverages (beer, wine, fortified wine and spirits) were included, 2) the wording of the third consumption item on heavy episodic drinking was changed from "one occasion" to "24 hours" to have a clear frame of reference, 3) alternative test items assessing the maximum drinking intake per 24 hours, and, 4) additional test items on the consumption of unrecorded alcohol were introduced to explore if they would add to the scale and improve the psychometric properties of the test in the Russian context [15, 33].

As the subsequent validation exercise of the AUDIT in different Russian regions has demonstrated, the introduced frequency-volume table that has been informed by previously developed materials as part of prior studies [39] has been probably the most relevant change in the now validated Russian version of the tool [33]. The table helps the health provider to convert the typical beverages and their volumes thar are reported by the patient into numbers of standard drinks and assists in the quantification of standard drinks and subsequent scoring and overall risk quantification. Along with the described challenges, the need for and usefulness of

such a table was identified as a possible solution to the issue of risk quantification as a concrete outcome of this exploratory study, similar to the need of changing the wording in the second test item from "once occasion" to "24 hours" to make this item understandable to patients and healthcare providers. Following the reported experience of one of the interviewed narcology doctors on using a colorful flashcard with pictures of beverages that represent standard drinks as part of another AUDIT study in Russia, the used flashcard design has also informed the final table as it is now combining both, pictures and textual information [33].

Besides the discussed practical aspects that have informed the subsequent validation process, the documented accounts provide deeper insights into aspects of Russian drinking patterns that might add to the understanding of the high alcohol-attributable burden of disease in Russia and ways of responding to these harms. For instance, the analyzed materials suggest that despite the various countermeasures introduced, several types of surrogate alcohols, as well as unrecorded alcohol in general, remain available in Russian communities, which is in line with the newest evidence on this topic [28, 42, 43]. Furthermore, this shows that Russia's national strategy to reduce alcohol consumption and the prevalence of alcohol use disorders at the population level might not have reached surrogate consumers as the most marginalized population group and that additional efforts are needed, specifically in the area of referral mechanisms between PHC and narcology services [16, 42, 44, 45]. Assessing unrecorded alcohol use and specifically consumption of surrogates as drinking behavior that is strongly associated with AUDs should therefore be considered when developing SBI programmes and materials in Russia and other former Soviet Union countries with similar drinking patterns and healthcare systems [46–48]. However, since surrogate alcohol use is perceived as a marker of severe forms of AUDs, it seems to not be cost-effective to include this into a general screening instrument for PHC facilities, but rather into specific addiction severity scales—a hypothesis that was later confirmed as part of the validation study of the Russian AUDIT [33].

## Conclusions

The findings from this study confirmed the clear need for an adapted country-specific version of the AUDIT to be used in different health settings in Russia for the early detection of potential AUDs and for initiating brief interventions as well as potential referrals.

The high level of alcohol-attributable harms remains one of the largest preventable public health issues in the Russian Federation despite the declining drinking levels after implementation of various alcohol control measures, such as price increases and restrictions of alcohol availability and marketing [49–53]. The provided analyses of drinking patterns in different patient groups might provide an explanation for this dynamic and suggests that the integration of SBI in the healthcare system is a key element in reducing alcohol-attributable harm in Russia and neighboring countries by bridging the gap between PHC and narcology services.

## Supporting information

**S1 Appendix. Standardized interview script for the qualitative expert interviews (to be delivered by the interviewer).**
(DOCX)

**S2 Appendix. Original Russian translation of the Alcohol Use Disorders Identification Test, using standard dinks (back-translated into English).**
(DOCX)

**S3 Appendix. Alternate updated version of the Russian translation of the Alcohol Use Disorders Identification Test, using a frequency-volume table (back-translated into English).**
(DOCX)

**S4 Appendix. Translated transcripts of the qualitative interviews used for the original analysis.**
(DOCX)

**S1 Table. Characteristics of the interviewed participants, according to sub-samples.**
(DOCX)

## Acknowledgments

We want to thank the patients and the healthcare professionals, who provided their invaluable input. Astrid Otto is thanked for referencing this manuscript. We also want to thank the Members of the RUS-AUDIT project advisory board: João Breda, Evgeny Bryun, Oxana Drapkina, Artyom Gil, Ruslan Khalfin, Daria Khaltourina, Tatiana Klimenko, Viktoria Madyanova, Oleg Salagay, Oleg Sonin, Kristina Soshkina and Konstantin Vyshinsky.

## Author Contributions

**Conceptualization:** Maria Neufeld, Carina Ferreira-Borges, Jürgen Rehm.

**Data curation:** Maria Neufeld, Jürgen Rehm.

**Formal analysis:** Maria Neufeld, Jürgen Rehm.

**Funding acquisition:** Carina Ferreira-Borges, Melita Vujnovic, Elena Yurasova.

**Investigation:** Maria Neufeld.

**Methodology:** Maria Neufeld, Jürgen Rehm.

**Project administration:** Maria Neufeld, Carina Ferreira-Borges, Elena Yurasova.

**Resources:** Maria Neufeld, Anna Bunova, Boris Gornyi, Eugenia Fadeeva, Evgenia Koshkina, Alexey Nadezhdin, Elena Tetenova.

**Software:** Maria Neufeld.

**Supervision:** Carina Ferreira-Borges, Jürgen Rehm.

**Validation:** Carina Ferreira-Borges, Jürgen Rehm.

**Visualization:** Maria Neufeld.

**Writing – original draft:** Maria Neufeld, Jürgen Rehm.

**Writing – review & editing:** Maria Neufeld, Carina Ferreira-Borges, Anna Bunova, Boris Gornyi, Eugenia Fadeeva, Evgenia Koshkina, Alexey Nadezhdin, Elena Tetenova, Melita Vujnovic, Elena Yurasova, Jürgen Rehm.

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
