## [Decision Letter · Decision Letter 0]

16 Mar 2022

PONE-D-21-18136Capturing Russian drinking patterns with the Alcohol Use Disorders Identification Test: an exploratory interview study in primary healthcare and narcology centers in MoscowPLOS ONE

Dear Dr. Neufeld,

Thank you for submitting your manuscript to PLOS ONE. After careful consideration, we feel that it has merit but does not fully meet PLOS ONE’s publication criteria as it currently stands. Therefore, we invite you to submit a revised version of the manuscript that addresses the points raised during the review process.

The manuscript has been evaluated by two reviewers, and their comments are available below.

The reviewers raise some overlapping concerns regarding the need for some additional methodological and statistical detail. In addition, they noted some copyediting errors and the need for some further descriptive information.

Could you please revise the manuscript to carefully address the concerns raised?

We look forward to receiving your revised manuscript.

Kind regards,

Avanti Dey, PhD

Staff Editor

PLOS ONE

Journal Requirements:

2. When reporting the results of qualitative research, we suggest consulting the COREQ guidelines: http://intqhc.oxfordjournals.org/content/19/6/349. In this case, please consider including more information on the number of interviewers, their training and characteristics. Moreover, please provide the interview guide used as a Supplementary File, in English and in the original language.

3. Thank you for stating the following in the Acknowledgments/ Funding Section of your manuscript: 

This work was supported through a grant of the Russian Government to the WHO European Office for the Prevention and Control of Noncommunicable Diseases. This funding source had no role in the design of the study, its execution, data collection and analyses, interpretation of the data, writing of the manuscript or decision to submit it.

This work was supported through a grant of the Russian Government to the WHO European Office for the Prevention and Control of Noncommunicable Diseases. This funding source had no role in the design of the study, its execution, data collection and analyses, interpretation of the data, writing of the manuscript or decision to submit it.

Reviewers' comments:

Reviewer's Responses to Questions

**Comments to the Author**

1. Is the manuscript technically sound, and do the data support the conclusions?

Reviewer #1: Yes

Reviewer #2: Yes

2. Has the statistical analysis been performed appropriately and rigorously? 

Reviewer #1: N/A

Reviewer #2: N/A

3. Have the authors made all data underlying the findings in their manuscript fully available?

Reviewer #1: No

Reviewer #2: Yes

4. Is the manuscript presented in an intelligible fashion and written in standard English?

Reviewer #1: Yes

Reviewer #2: Yes

5. Review Comments to the Author

Reviewer #1: This is an important subject area because the alcohol content in a standard drink in different countries varies and it is necessary to adapt that audit questionnaire to local conditions. It is difficult in Russia were the concept of standard drink not exist and even more difficult is the definition of drinking occasion. The first aim of this study was to identify problems with understanding all of the audit items. The second aim was to improve the clarity of those items. A third aim was to explore possible solutions and the existing practices in screening at health care providers. A fourth aim was to integrate experiences of the use of the audit questionnaire. However I have some comments I hope will make the text easier to follow.In table 1 information should be written in the usual way, now it seems not to be a table. Maybe it is necessary with a statistical test between the groups regarding the age. I would be happy to see the citations from the interviews because they are the data which the conclusions are built upon. Now the conclusions standing alone which makes it difficult to control the conclusions for the reader. It can be problematic to Translate the concept standard drink into doses without specifying what a dose is and how much alcohol is contains. It is not clear how the volumes are calculated from standard portions in table 4. Item 3 in the audit Define if the person is a binge drinker or not. But in the Russian version according to table five item three assess maximal amount alcohol a person drinks in a day and how often. It is not comparable with binge drinking frequency. Another problem with measuring alcohol consumption in Russia is there a high number of people who drink unrecorded alcohol that is not assessed with audit.

Reviewer #2: The article is very well written and very clear, the study was needed and the adaptations will be useful in research and clinical practice, as planned. A few typos remain, so another look would be helpful (Background section: 'patters' (patterns); ...creation and validation for (instead of of) a Russian version of...

I also suggest to add the sex distribution of the health care professionals interviewed, the only group without that information.

6. PLOS authors have the option to publish the peer review history of their article (what does this mean?). If published, this will include your full peer review and any attached files.

Reviewer #1: **Yes: **Håkan Källmen

Reviewer #2: **Yes: **Maristela Goldnadel Monteiro

---

## [Author Response · Author response to Decision Letter 0]

12 Jun 2022

Reviewer #1: This is an important subject area because the alcohol content in a standard drink in different countries varies and it is necessary to adapt that audit questionnaire to local conditions. It is difficult in Russia were the concept of standard drink not exist and even more difficult is the definition of drinking occasion. The first aim of this study was to identify problems with understanding all of the audit items. The second aim was to improve the clarity of those items. A third aim was to explore possible solutions and the existing practices in screening at health care providers. A fourth aim was to integrate experiences of the use of the audit questionnaire. However I have some comments I hope will make the text easier to follow.

1. In table 1 information should be written in the usual way, now it seems not to be a table. Maybe it is necessary with a statistical test between the groups regarding the age. 

A1: We have deliberately provided this information in a table form to give readers at least a rough idea of the different subsamples and the interviewees, who were recruited using a convenience sampling method, as discussed in the limitations section. It does not seem sensible to apply a statistical test to the different groups regarding age given that this is a very small non-representative sample. We also have tried rewriting this section as advised, putting the information contained into text format, but it made the description of the samples even more complicated and less transparent. Therefore, we would like to continue to use the current format (Table 1). If deemed necessary, we can move the table to the Web Appendix, but describing all the information that it contains in written form within the text does not seem to improve readability of the manuscript.

2. I would be happy to see the citations from the interviews because they are the data which the conclusions are built upon. Now the conclusions standing alone which makes it difficult to control the conclusions for the reader. 

A2: We have now added several citations into the different sections of the manuscript, and we hope that they provide a clearer idea of how we have arrived at the conclusions.

3. It can be problematic to Translate the concept standard drink into doses without specifying what a dose is and how much alcohol is contains. It is not clear how the volumes are calculated from standard portions in table 4. 

A3: We have clarified in the manuscript (in relation to Table 3 and Table 4) that these items are taken from the two different Russian translations, which are featured on the AUDIT website of John Saunders (https://auditscreen.org/translations/). We have identified over 60 different Russian translations of the AUDIT and, in many of them, it was not clear how the standard drinks or the final scores were calculated and there were many other, more serios issues identified (Bunova et al., 2021; Neufeld et al., 2021). 

The main point of the exercise was to explore how PHC and narcology patients perceive the second AUDIT item when its content is represented in different way. We were not measuring the patient’s actual alcohol consumption or related risks.

References:

Bunova, A., Neufeld, M., Ferreira-Borges, C., Bryun, E., Fadeeva, E., Gil, A., ... & Rehm, J. (2021). The Russian translations of the Alcohol Use Disorders Identification Test (AUDIT): A document analysis and discussion of implementation challenges. The Intezrnational Journal of Alcohol and Drug Research, 9(1), 20-29.

Neufeld, M., Bunova, A., Ferreira-Borges, C., Bryun, E., Fadeeva, E., Gil, A., ... & Rehm, J. (2021). The Alcohol Use Disorders Identification Test (AUDIT) in the Russian language-a systematic review of validation efforts and application challenges. Substance abuse treatment, prevention, and policy, 16(1), 1-14. 

Auditscreen.org. AUDIT 2021 Available from: https://auditscreen.org/cmsb/uploads/audit-russian.pdf.Accessed: 26/04/2021 53. 

Auditscreen.org. AUDIT (Russian): Приложение C 2021 Available from: https://auditscreen.org/cmsb/uploads/audit-russian-new.pdf Accessed: 26/04/2021

4. Item 3 in the audit Define if the person is a binge drinker or not. But in the Russian version according to table five item three assess maximal amount alcohol a person drinks in a day and how often. It is not comparable with binge drinking frequency. Another problem with measuring alcohol consumption in Russia is there a high number of people who drink unrecorded alcohol that is not assessed with audit.

A4: As described in this section, patients and healthcare professionals alike reported difficulties in understanding the concept of binge drinking and what was meant by the term “one occasion”. Because of this issue, the interviewer was exploring what “one occasion” meant to different interviewees and, if difficulties in reporting arose, the interviewer clarified that respondents should consider an occasion as being equivalent to a 24-hour period since this was a concept well understood by everyone. This part of the interview was deliberately left open-ended and exploratory in nature to capture the different perceptions and cognitive concepts, as described in detail on pp. 22-24. 

The same can be said for unrecorded alcohol, which was included in this exploratory qualitative study as an important concept exactly because this is commonly consumed in Russia but not covered by the AUDIT. The results discussed in this article were used in the empirical validation study of the AUDIT in different Russian regions, where specific questions unrecorded was included as well, but the validation results have shown that adding specific items for unrecorded alcohol does not improve the performance of the tool. This is, however, in line with the present results that consumption of unrecorded alcohol is closely linked with alcohol dependence, suggesting that it is worth exploring the inclusion of such items into a Russia-specific addiction scale. 

Reviewer #2: The article is very well written and very clear, the study was needed and the adaptations will be useful in research and clinical practice, as planned. A few typos remain, so another look would be helpful (Background section: 'patters' (patterns); ...creation and validation for (instead of of) a Russian version of...

A1. The manuscript was again proofread by a native speaker and we hope that we have addressed all typos and other issues.

We thank the reviewers for their comments and hope that we were able to address all of them. The editing of the initial manuscript has led to an increase in the word count because of the additional original interview quotes requested by Reviewer 1. We hope the editor agrees that these measures have led to a clearer and more readable manuscript. 

For the authors, 

Maria Neufeld

---

## [Editor Report · Decision Letter 1]

28 Jun 2022

PONE-D-21-18136R1Capturing Russian drinking patterns with the Alcohol Use Disorders Identification Test: an exploratory interview study in primary healthcare and narcology centers in MoscowPLOS ONE

Dear Dr. Neufeld,

Thank you for submitting your manuscript to PLOS ONE. After careful consideration, we feel that it has merit but does not fully meet PLOS ONE’s publication criteria as it currently stands. Therefore, we invite you to submit a revised version of the manuscript that addresses the points raised during the review process.

We look forward to receiving your revised manuscript.

Kind regards,

Håkan Källmén

Guest Editor

PLOS ONE

Journal Requirements:

Additional Editor Comments (if provided):

As I have earlier revised this manuscript I recognize that more citations from interviews are added that makes it easier to follow. The first purpose of the study was to highlight problems that exist with the understanding of the issues in Audit. It is quite clear from the study that the major problems with understanding apply to the concepts of standard drink and also drinking occasion. Thus, the first purpose is achieved. The authors also propose solutions to adapt the issues to the Russian alcohol culture, which was the second purpose.

Whether the third and fourth objectives are achieved is a little more unclear as the author does not discuss different practical solutions for screening and integrating different experiences of Audit.

The tables in the manuscript are not tables but information sheets. Please change the that

I suggest accept after considering this
---

## [Author Response · Author response to Decision Letter 1]

16 Aug 2022

Dear editor,

Thank you very much for the careful consideration of the manuscript. 

We have included an additional paragraph that discusses the findings of the study in relation to the third and fourth stated objective in greater detail, i.e. the conversion table as a practical solution and its integration with a visual flashcard depicting different beverages as standard drinks (pp. 34-35).We hope that the important implications that this exploratory study has had on the subsequent validation study of the AUDIT in the Russian Federation are now clearer.

We have now also revised the paragraph with the sample description (p.14), following earlier reviewer feedback, and have provided the sample size characteristics in-text, thereby removing Table 1 from the manuscript. However, in the current submission guidelines of the journal I could not find any guidance on how information sheets should be handled in the manuscript, following your suggestion to change the tables into information sheet. Please kindly advise on how this should be handled, and I will make the necessary changes within the next 24 hours. 

Thank you for your considerate feedback.

With best wishes

---

## [Editor Report · Decision Letter 2]

19 Aug 2022

PONE-D-21-18136R2Capturing Russian drinking patterns with the Alcohol Use Disorders Identification Test: an exploratory interview study in primary healthcare and narcology centers in MoscowPLOS ONE

Dear Dr. Neufeld,

Thank you for submitting your manuscript to PLOS ONE. After careful consideration, we feel that it has merit but does not fully meet PLOS ONE’s publication criteria as it currently stands. Therefore, we invite you to submit a revised version of the manuscript that addresses the points raised during the review process.

We look forward to receiving your revised manuscript.

Kind regards,

Håkan Källmén

Guest Editor

PLOS ONE

Journal Requirements:

Additional Editor Comments (if provided):

Dear authors I think it is not correct to use the word table in a qualitative study. I think it is better to delete the words Table 1, Table 2, Table 3 and Table 4. Only the explaining text should remain. After those changes i suggest an acceptance of the manuscript
---

## [Author Response · Author response to Decision Letter 2]

19 Aug 2022

Dear editor,

Thank you very much for your considerate feedback. We have now made the suggested changes.

With best wishes

Maria Neufeld (for the authors)

---

## [Editor Report · Decision Letter 3]

24 Aug 2022

Capturing Russian drinking patterns with the Alcohol Use Disorders Identification Test: an exploratory interview study in primary healthcare and narcology centers in Moscow

PONE-D-21-18136R3

Dear Dr. Neufeld,

We’re pleased to inform you that your manuscript has been judged scientifically suitable for publication and will be formally accepted for publication once it meets all outstanding technical requirements.

Kind regards,

Håkan Källmén

Guest Editor

PLOS ONE

Additional Editor Comments (optional):

Congratulations. This important manuscript has improved and should be published